# A Neural Dirichlet Process Mixture Model for Task-Free Continual Learning

**Soochan Lee, Junsoo Ha, Dongsu Zhang & Gunhee Kim**
Department of Computer Science, Seoul National University, Seoul, Republic of Korea
`{soochan.lee,junsoo.ha}@vision.snu.ac.kr,{96lives,gunhee}@snu.ac.kr`
`http://vision.snu.ac.kr/projects/cn-dpm`

## Abstract

Despite the growing interest in continual learning, most of its contemporary works have been studied in a rather restricted setting where tasks are clearly distinguishable, and task boundaries are known during training. However, if our goal is to develop an algorithm that learns as humans do, this setting is far from realistic, and it is essential to develop a methodology that works in a *task-free* manner. Meanwhile, among several branches of continual learning, expansion-based methods have the advantage of eliminating catastrophic forgetting by allocating new resources to learn new data. In this work, we propose an expansion-based approach for task-free continual learning. Our model, named *Continual Neural Dirichlet Process Mixture* (CN-DPM), consists of a set of neural network experts that are in charge of a subset of the data. CN-DPM expands the number of experts in a principled way under the Bayesian nonparametric framework. With extensive experiments, we show that our model successfully performs task-free continual learning for both discriminative and generative tasks such as image classification and image generation.

## 1 Introduction

Humans consistently encounter new information throughout their lifetime. The way the information is provided, however, is vastly different from that of conventional deep learning where each mini-batch is iid-sampled from the whole dataset. Data points adjacent in time can be highly correlated, and the overall distribution of the data can shift drastically as the training progresses. *Continual learning* (CL) aims at imitating incredible human's ability to learn from a non-iid stream of data without catastrophically forgetting the previously learned knowledge.

Most CL approaches (Aljundi et al., 2018; 2017; Lopez-Paz & Ranzato, 2017; Kirkpatrick et al., 2017; Rusu et al., 2016; Shin et al., 2017; Yoon et al., 2018) assume that the data stream is explicitly divided into a sequence of tasks that are known at training time. Since this assumption is far from realistic, *task-free* CL is more practical and demanding but has been largely understudied with only a few exceptions of (Aljundi et al., 2019a;b). In this general CL, not only is explicit task definition unavailable but also the data distribution gradually shifts without a clear task boundary.

Meanwhile, existing CL methods can be classified into three different categories (Parisi et al., 2019): regularization, replay, and expansion methods. Regularization and replay approaches address the catastrophic forgetting by regularizing the update of a specific set of weights or replaying the previously seen data, respectively. On the other hand, the expansion methods are different from the two approaches in that it can expand the model architecture to accommodate new data instead of fixing it beforehand. Therefore, the expansion methods can bypass catastrophic forgetting by preventing pre-existing components from being overwritten by the new information. The critical limitation of prior expansion methods, however, is that the decisions of when to expand and which resource to use heavily rely on explicitly given task definition and heuristics.

In this work, our goal is to propose a novel expansion-based approach for task-free CL. Inspired by the Mixture of Experts (MoE) (Jacobs et al., 1991), our model consists of a set of *experts*, each of which is in charge of a subset of the data in a stream. The model expansion (i.e., adding more

experts) is governed by the *Bayesian nonparametric* framework, which determines the model complexity by the data, as opposed to the parametric methods that fix the model complexity before training. We formulate the task-free CL as an online variational inference of Dirichlet process mixture models consisting of a set of neural experts; thus, we name our approach as the *Continual Neural Dirichlet Process Mixture* (CN-DPM) model.

We highlight the key contributions of this work as follows.

- We are one of the first to propose an expansion-based approach for task-free CL. Hence, our model not only prevents catastrophic forgetting but also applies to the setting where no task definition and boundaries are given at both training and test time. Our model named CN-DPM consists of a set of neural network experts, which are expanded in a principled way built upon the Bayesian nonparametrics that have not been adopted in general CL research.

- Our model can deal with both generative and discriminative tasks of CL. With several benchmark experiments of CL literature on MNIST, SVHN, and CIFAR 10/100, we show that our model successfully performs multiple types of CL tasks, including image classification and generation.

## 2 BACKGROUND AND RELATED WORK

### 2.1 CONTINUAL LEARNING

Parisi et al. (2019) classify CL approaches into three branches: regularization (Kirkpatrick et al., 2017; Aljundi et al., 2018), replay (Shin et al., 2017) and expansion (Aljundi et al., 2017; Rusu et al., 2016; Yoon et al., 2018) methods. Regularization and replay approaches fix the model architecture before training and prevent catastrophic forgetting by regularizing the change of a specific set of weights or replaying previously learned data. Hybrids of replay and regularization also exist, such as Gradient Episodic Memory (GEM) (Lopez-Paz & Ranzato, 2017; Chaudhry et al., 2019a). On the other hand, methods based on expansion add new network components to learn new data. Conceptually, such direction has the following advantages compared to the first two: (i) catastrophic forgetting can be eliminated since new information is not overwritten on pre-existing components and (ii) the model capacity is determined adaptively depending on the data.

**Task-Free Continual Learning**. All the works mentioned above heavily rely on explicit task definition. However, in real-world scenarios, task definition is rarely given at training time. Moreover, the data domain may gradually shift without any clear task boundary. Despite its importance, task-free CL has been largely understudied; to the best of our knowledge, there are only a few works (Aljundi et al., 2019a;b; Rao et al., 2019), each of which is respectively based on regularization, replay, and a hybrid of replay and expansion. Specifically, Aljundi et al. (2019a) extend MAS (Aljundi et al., 2018) by adding heuristics to determine when to update the importance weights with no task definition. In their following work (Aljundi et al., 2019b), they improve the memory management algorithm of GEM (Lopez-Paz & Ranzato, 2017) such that the memory elements are carefully selected to minimize catastrophic forgetting. While focused on unsupervised learning, Rao et al. (2019) is a parallel work that shares several similarities with our method, e.g., model expansion and short-term memory. However, due to their model architecture, expansion is not enough to stop catastrophic forgetting; consequently, generative replay plays a crucial role in Rao et al. (2019). As such, it can be categorized as a hybrid of replay and expansion. More detailed comparison between our method and Rao et al. (2019) is deferred to Appendix M.

### 2.2 DIRICHLET PROCESS MIXTURE MODELS

We briefly review the Dirichlet process mixture (DPM) model (Antoniak, 1974; Ferguson, 1983), and a variational method to approximate the posterior of DPM models in an online setting: Sequential Variational Approximation (SVA) (Lin, 2013). For a more detailed review, refer to Appendix A.

**Dirichlet Process Mixture (DPM)**. The DPM model is often applied to clustering problems where the number of clusters is not known in advance. The generative process of a DPM model is

$$x_n \sim p(x; \theta_n), \quad \theta_n \sim G, \quad G \sim \mathrm{DP}(\alpha, G_0), \tag{1}$$

where $x_n$ is the $n$-th data, and $\theta_n$ is the $n$-th latent variable sampled from $G$, which itself is a distribution sampled from a Dirichlet process (DP). The DP is parameterized by a concentration parameter $\alpha$ and a base distribution $G_0$. The expected number of clusters is proportional to $\alpha$, and $G_0$ is the marginal distribution of $\theta$ when $G$ is marginalized out. Since $G$ is discrete with probability 1 (Teh, 2010), same values can be sampled multiple times for $\theta$. If $\theta_n = \theta_m$, the two data points $x_n$ and $x_m$ belong to the same cluster. An alternative formulation uses the variable $z_n$ that indicates to which cluster the $n$-th data belongs such that $\theta_n = \phi_{z_n}$ where $\phi_k$ is the parameter of the $k$-th cluster. In the context of this paper, $\phi_k$ refers to the parameters of the $k$-th expert.

**Approximation of the Posterior of DPM Models**. Since the exact inference of the posterior of DPM models is infeasible, approximate inference methods are applied. Among many approximation methods, we adopt the Sequential Variational Approximation (SVA) (Lin, 2013). While the data is given one by one, SVA sequentially determines $\rho_n$ and $\nu_k$, which are the variational approximation for the distribution of $z_n$ and $\phi_k$ respectively. Since $\rho_n$ satisfies $\sum_k \rho_{n,k} = 1$ and $\rho_{n,k} >= 0$, $\rho_{n,k}$ can be interpreted as the probability of $n$-th data belonging to $k$-cluster and is often called responsibility. $\rho_{n+1}$ and $\nu^{(n+1)}$ at step $n+1$ are computed as:

$$\rho_{n+1,k} \propto \begin{cases} (\sum_{i=1}^n \rho_{i,k}) \int_\phi p(x_{n+1}|\phi)\nu_k^{(n)}(d\phi) & \text{if } 1 \le k \le K \\ \alpha \int_\phi p(x_{n+1}|\phi)G_0(d\phi) & \text{if } k = K+1 \end{cases}, \tag{2}$$

$$\nu_k^{(n+1)}(d\phi) \propto \begin{cases} G_0(d\phi) \prod_{i=1}^{n+1} p(x_i|\phi)^{\rho_{i,k}} & \text{if } 1 \le k \le K \\ G_0(d\phi) p(x_{n+1}|\phi)^{\rho_{n+1,k}} & \text{if } k = K+1 \end{cases}. \tag{3}$$

In practice, SVA adds a new component only when $\rho_{K+1}$ is greater than a certain threshold $\epsilon$. If $G_0$ and $p(x_i|\phi)$ are not a conjugate pair, stochastic gradient descent (SGD) is used to find the MAP estimation $\hat{\phi}$ with a learning rate of $\lambda$ instead of calculating the whole distribution $\nu_k$:

$$\hat{\phi}_k^{(n+1)} \leftarrow \hat{\phi}_k^{(n)} + \lambda(\nabla_{\hat{\phi}_k^{(n)}} \log G_0(\hat{\phi}_k^{(n)}) + \nabla_{\hat{\phi}_k^{(n)}} \log p(x|\hat{\phi}_k^{(n)})). \tag{4}$$

**DPM for Discriminative Tasks**. DPM can be extended to discriminative tasks where each data point is an input-output pair $(x, y)$, and the goal is to learn the conditional distribution $p(y|x)$. To use DPM, which is a generative model, for discriminative tasks, we first learn the joint distribution $p(x, y)$ and induce the conditional distribution from it: $p(y|x) = p(x, y) / \int_y p(x, y)$. The joint distribution modeled by each component can be decomposed as $p(x, y|z) = p(y|x, z)p(x|z)$ (Rasmussen & Ghahramani, 2002; Shahbaba & Neal, 2009).

**DPM in Related Fields**. Recent works of Nagabandi et al. (2019) and Jerfel et al. (2019) exploit the DPM framework to add new components without supervision in the meta-learning context. Nagabandi et al. (2019) apply DPM to the model-based reinforcement learning to predict the next state from a given state-action pair. When a new task appears, they add a component under the DPM framework to handle predictions in the new task. Jerfel et al. (2019) apply DPM to online meta-learning. Extending MAML (Finn et al., 2017), they assume that similar tasks can be grouped into a super-task in which the parameter initialization is shared among tasks. DPM is exploited to find the super-tasks and the parameter initialization for each super-task. Therefore, it can be regarded as a meta-level CL method. These works, however, lack generative components, which are often essential to infer the responsible component at test time, as will be described in the next section. As a consequence, it is not straightforward to extend their algorithms to other CL settings beyond model-based RL or meta-learning. In contrast, our method implements a DPM model that is applicable to general task-free CL.

## 3 APPROACH

We aim at general *task-free* CL, where the number of tasks and task descriptions are not available at both training and test time. We even consider the case where the data stream cannot be split into

separate tasks in Appendix F. All of the existing expansion methods are not task-free since they require task definition at training (Aljundi et al., 2017) or even at test time (Rusu et al., 2016; Xu & Zhu, 2018; Li et al., 2019). We propose a novel expansion method that automatically determines when to expand and which component to use. We first deal with generative tasks and generalize them into discriminative ones.

### 3.1 CONTINUAL LEARNING AS MODELING OF THE MIXTURE DISTRIBUTION

We can formulate a CL scenario as a stream of data involving different tasks $\mathcal{D}_1, \mathcal{D}_2, ...$ where each task $\mathcal{D}_k$ is a set of data sampled from a (possibly) distinct distribution $p(x|z = k)$. If $K$ tasks are given so far, the overall distribution is expressed as the mixture distribution:

$$p(x) = \sum_{k=1}^{K} p(x|z = k)p(z = k), \tag{5}$$

where $p(z = k)$ can be approximated by $N_k/N$ where $N_k = |\mathcal{D}_k|$ and $N = \sum_k N_k$. The goal of CL is to learn the mixture distribution in an online manner. Regularization and replay methods directly model the approximate distribution $p(x; \phi)$ parameterized by a single component $\phi$ and update it to fit the overall distribution $p(x)$. When updating $\phi$, however, they do not have full access to all the previous data, and thus the information of previous tasks is at risk of being lost as more tasks are learned. Another way to solve CL is to use a mixture model: approximating each $p(x|z = k)$ with $p(x; \phi_k)$. If we learn a new task distribution $p(x|z = K + 1)$ with new parameter $\phi_{K+1}$ and leave the existing parameters intact, we can preserve the knowledge of the previous tasks. The expansion-based CL methods follow this idea.

Similarly, in the discriminative task, the goal of CL is to model the overall conditional distribution, which is a mixture of task-wise conditional distribution $p(y|x, z = k)$:

$$p(y|x) = \sum_{k=1}^{K} p(y|x, z = k)p(z = k|x). \tag{6}$$

Prior expansion methods use expert networks each of which models a task-wise conditional distribution $p(y|x; \phi_k)$[1]. However, a new problem arises in expansion methods: choosing the right expert given $x$, i.e., $p(z|x)$ in Eq.(6). Existing methods assume that explicit task descriptor $z$ is given, which is generally not true in human-like learning scenarios. That is, we need a *gating mechanism* that can infer $p(z|x)$ only from $x$ (i.e., which expert should process $x$). With the gating, the model prediction naturally reduces to the sum of expert outputs weighted by the gate values, which is the mixture of experts (MoE) (Jacobs et al., 1991) formulation: $p(y|x) \approx \sum_k p(y|x; \phi_k)p(z = k|x)$.

However, it is not possible to use a single gate network as in Shazeer et al. (2017) to model $p(z|x)$ in CL; since the gate network is a classifier that finds the correct expert for a given data, training it in an online setting causes catastrophic forgetting. Thus, one possible solution to replace a gating network is to couple each expert $k$ with a generative model that represents $p(x|z = k)$ as in Rasmussen & Ghahramani (2002) and Shahbaba & Neal (2009). As a result, we can build a gating mechanism without catastrophic forgetting as

$$p(y|x) \approx \sum_k p(y|x; \phi_k^D)p(z = k|x) \approx \sum_k p(y|x; \phi_k^D)\frac{p(x; \phi_k^G)p(z = k)}{\sum_{k'} p(x; \phi_{k'}^G)p(z = k')}, \tag{7}$$

where $p(z = k) \approx N_k/N$. We also differentiate the notation for the parameters of discriminative models for classification and generative models for gating by the superscript $D$ and $G$.

If we know the true assignment of $z$, which is the case of task-based CL, we can independently train a discriminative model (i.e., $p(y|x; \phi_k^D)$) and a generative model (i.e., $p(x; \phi_k^G)$) for each task $k$. In task-free CL, however, $z$ is unknown, so the model needs to infer the posterior $p(z|x, y)$. Even worse, the total number of experts is unknown beforehand. Therefore, we propose to employ a Bayesian nonparametric framework, specifically the Dirichlet process mixture (DPM) model, which can fit a mixture distribution with no prefixed number of components. We use SVA described in

---

[1] The models with multiple output heads sharing the same base network (Rusu et al., 2016; Yoon et al., 2018) can also fall into this category as the expert correspond to each subnetwork connected to an output head.

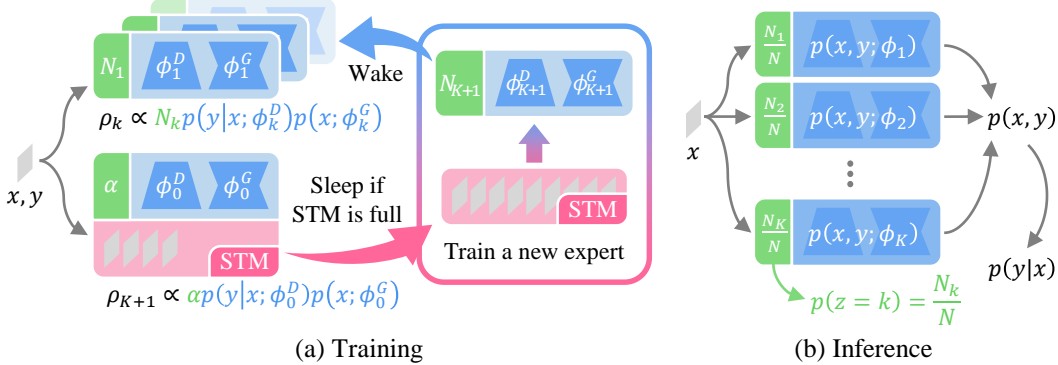

(a) Training            (b) Inference

Figure 1: Overview of our CN-DPM model. Each expert $k$ (blue boxes) contains a discriminative component for modeling $p(y|x; \phi_k^D)$ and a generative component for modeling $p(x; \phi_k^G)$, jointly representing $p(x, y; \phi_k)$. We also keep the assigned data count $N_k$ per expert. (a) During training, each sample $(x, y)$ coming in a sequence is evaluated by every expert to calculate the responsibility $\rho_k$ of each expert. If $\rho_{K+1}$ is high enough, i.e., none of the existing experts is responsible, the data is stored into short-term memory (STM). Otherwise, it is learned by the corresponding expert. When STM is full, a new expert is created from the data in STM. (b) Since CN-DPM is a generative model, we first compute the joint distribution $p(x, y)$ for a given $x$, from which it is trivial to infer $p(y|x)$.

section 2.2 to approximate the posterior in an online setting. Although SVA is originally designed for the generative tasks, it is easily applicable to discriminative tasks by making each component $k$ to model $p(x, y|z) = p(y|x, z)p(x|z)$.

## 3.2 THE CONTINUAL NEURAL DIRICHLET PROCESS MIXTURE (CN-DPM) MODEL

The proposed approach for task-free CL, named *Continual Neural Dirichlet Process Mixture* (CN-DPM) model, consists of a set of *experts*, each of which is associated with a discriminative model (classifier) and a generative model (density estimator). More specifically, the classifier models $p(y|x, z = k)$, for which we can adopt any classifier or regressor using deep neural networks, while the density estimator describes the marginal likelihood $p(x|z = k)$, for which we can use any explicit density model such as VAEs (Kingma & Welling, 2014) and PixelRNN (Oord et al., 2016). We respectively denote the classifier and the density estimator of expert $k$ as $p(y|x; \phi_k^D)$ and $p(x; \phi_k^G)$, where $\phi_k^D$ and $\phi_k^G$ are the parameters of the models. Finally, the prediction $p(y|x)$ can be obtained from Eq.(7) by plugging in the output of the classifier and the density estimator. Note that the number of experts is not prefixed but expanded via the DPM framework. Figure 1 illustrates the overall training and inference process of our model.

**Training**. We assume that samples sequentially arrive one at a time during training. For a new sample, we first decide whether the sample should be assigned to an existing expert or a new expert should be created for it. Suppose that samples up to $(x_n, y_n)$ are sequentially processed and $K$ experts are already created when a new sample $(x_{n+1}, y_{n+1})$ arrives. We compute the responsibility $\rho_{n+1,k}$ as follows:

$$\rho_{n+1,k} \propto \begin{cases} (\sum_{i=1}^n \rho_{i,k}) \, p(y_{n+1}|x_{n+1}; \hat{\phi}_k^D)p(x_{n+1}; \hat{\phi}_k^G) & \text{if } 1 \le k \le K \\ \alpha p(y_{n+1}|x_{n+1}; \hat{\phi}_0^D)p(x_{n+1}; \hat{\phi}_0^G) \text{ where } \hat{\phi}_0 \sim G_0(\phi) & \text{if } k = K+1 \end{cases} \tag{8}$$

where $G_0$ is a distribution corresponding to the weight initialization. If $\arg \max_k \rho_{n+1,k} \ne K + 1$, the sample is assigned to the existing experts proportional to $\rho_{n+1,k}$, and the parameters of the experts are updated with the new sample by Eq.(4) such that $\hat{\phi}_k$ is the MAP approximation given the data assigned up to the current time step. Otherwise, we create a new expert.

**Short-Term Memory**. However, it is not a good idea to create a new expert immediately and initialize it to be the MAP estimation given $x_{n+1}$. Since both the classifier and density estimator of an expert are neural networks, training the new expert with only a single example leads to severe overfitting. To mitigate this issue, we employ *short-term memory* (STM) to collect sufficient data

before creating a new expert. When a data point is classified as new, we store it to the STM. Once the STM reaches its maximum capacity $M$, we stop the data inflow for a while and train a new expert with the data in the STM for multiple epochs until convergence. We call this procedure *sleep phase*. After sleep, the STM is emptied, and the newly trained expert is added to the expert pool. During the subsequent wake phase, the expert is learned from the data assigned to it. This STM trick assumes that the data in the STM belong to the same expert. We empirically find that this assumption is acceptable in many CL settings where adjacent data are highly correlated. The overall training procedure is described in Algorithm 1. Note that we use $\rho_{n,0}$ instead of $\rho_{n,K+1}$ in the algorithm for brevity.

**Inference**. At test time, we infer $p(y|x)$ from the collaboration of the learned experts as in Eq.(7).

**Techniques for Practicality**. Naively adding a new expert has two major problems: (i) the number of parameters grows unnecessarily large as the experts redundantly learn common features and (ii) there is no positive transfer of knowledge between experts. Therefore, we propose a simple method to share parameters between experts. When creating a new expert, we add lateral connections to the features of the previous experts similar to Rusu et al. (2016). To prevent catastrophic forgetting in the existing experts, we block the gradient from the new expert. In this way, we can greatly reduce the number of parameters while allowing positive knowledge transfer. More techniques such as sparse regularization in Yoon et al. (2018) can be employed to reduce redundant parameters further. As they are orthogonal to our approach, we do not use such techniques in our experiments. Another effective technique that we use in the classification experiments is adding a temperature parameter to the classifier. Since the range of $\log p(x|z)$ is far broader than $\log p(y|x,z)$, the classifier has almost no effect without proper scaling. Thus, we can increase overall accuracy by adjusting the relative importance of images and labels. We also introduce an algorithm to prune redundant experts in Appendix D, and discuss further practical issues of CN-DPM in Appendix B.

---

**Algorithm 1** Training of the Continual Neural Dirichlet Process Mixture (CD-NDP) Model

---

**Require:** Data $(x_1, y_1), ..., (x_N, y_N)$, concentration $\alpha$, base measure $G_0$, short-term memory capacity $M$, learning rate $\lambda$
1: $\mathcal{M} \leftarrow \emptyset$ {Short-term memory}
2: $K \leftarrow 0$ {Number of experts}
3: $N_0 \leftarrow \alpha$; $\hat{\phi}_0 \leftarrow \text{Sample}(G_0)$
4: **for** $n = 1$ **to** $N$ **do**
5:  **for** $k = 0$ **to** $K$ **do**
6:   $l_k \leftarrow p(y_n|x_n; \hat{\phi}_k^D)p(x_n; \hat{\phi}_k^G)$
7:   $\rho_{n,k} \leftarrow N_k l_k$
8:  **end for**
9:  $\rho_{n,0:K} \leftarrow \rho_{n,0:K} / \sum_{k=0}^{K} \rho_{n,k}$
10:  **if** $\arg\max_k \rho_{n,k} = 0$ **then**
11:   {Save $x_n$ to short-term memory}
12:    $\mathcal{M} \leftarrow \{x_n\} \cup \mathcal{M}$
13:    **if** $|\mathcal{M}| \geq M$ **then** {Add new expert}
14:     $\hat{\phi}_{K+1} \leftarrow \text{FindMAP}(\mathcal{M}, G_0)$
15:     $N_{K+1} \leftarrow |\mathcal{M}|$; $\mathcal{M} \leftarrow \emptyset$
16:     $K \leftarrow K + 1$
17:    **end if**
18:   **else** {Update existing experts}
19:    $\rho_{n,1:K} \leftarrow \rho_{n,1:K} / \sum_{k=1}^{K} \rho_{n,k}$
20:    **for** $k = 1$ **to** $K$ **do**
21:     $N_k \leftarrow N_k + \rho_{n,k}$
22:     $\hat{\phi}_k \leftarrow \hat{\phi}_k + \rho_{n,k} \lambda \nabla_{\hat{\phi}_k} \log l_k$
23:    **end for**
24:   **end if**
25: **end for**

---

## 4 EXPERIMENTS

We evaluate the proposed CN-DPM model in task-free CL with four benchmark datasets. Appendices include more detailed model architecture, additional experiments, and analyses.

### 4.1 CONTINUAL LEARNING SCENARIOS

A CL scenario defines a sequence of *tasks* where the data distribution for each task is assumed to be different from others. Below we describe the task-free CL scenarios used in the experiments. At both train and test time, the model cannot access the task information. Unless stated otherwise, each task is presented for a single epoch (i.e., a completely online setting) with a batch size of 10.

**Split-MNIST** (Zenke et al., 2017). The MNIST dataset (LeCun et al., 1998) is split into five tasks, each containing approximately 12K images of two classes, namely (0/1, 2/3, 4/5, 6/7, 8/9). We conduct both classification and generation experiments in this scenario.

Table 1: Test scores and the numbers of parameters in task-free CL on Split-MNIST, MNIST-SVHN, and Split-CIFAR100 scenarios. Note that iid-∗ baselines are not CL methods.

| Method | Split-MNIST | | Split-MNIST (Gen.) | | MNIST-SVHN | | Split-CIFAR100 | |
|---|---|---|---|---|---|---|---|---|
| | Acc. (%) | Param. | bits/dim | Param. | Acc. (%) | Param. | Acc. (%) | Param. |
| iid-offline | 98.63 | 478K | 0.1806 | 988K | 96.69 | 11.2M | 73.80 | 11.2M |
| iid-online | 96.18 | 478K | 0.2156 | 988K | 95.24 | 11.2M | 20.46 | 11.2M |
| Fine-tune | 19.43 | 478K | 0.2817 | 988K | 83.35 | 11.2M | 2.43 | 11.2M |
| Reservoir | 85.69 | 478K | 0.2234 | 988K | 94.12 | 11.2M | 10.01 | 11.2M |
| CN-DPM | **93.23** | 524K | **0.2110** | 970K | **94.46** | 7.80M | **20.10** | 19.2M |

Table 2: Performance comparison on Split-CIFAR10 with various scenario length.

| Method | Split-CIFAR10 Acc. (%) | | | Param. |
|---|---|---|---|---|
| | 0.2 Epoch | 1 Epoch | 10 Epochs | |
| iid-offline | 93.17 | 93.17 | 93.17 | 11.2M |
| iid-online | 36.65 | 62.79 | 83.19 | 11.2M |
| Fine-tune | 12.68 | 18.08 | 19.31 | 11.2M |
| Reservoir | 37.09 | 44.00 | 43.82 | 11.2M |
| GSS | 33.56 | – | – | 11.2M |
| CN-DPM | **41.78** | **45.21** | **46.98** | 4.60M |

Table 3: Dissecting the performance of CN-DPM.

| Acc. Type | Split-CIFAR10 | Split-CIFAR100 |
|---|---|---|
| Classifier (init) | 88.20 | 55.42 |
| Classifier (final) | 88.20 | 55.24 |
| Gating (VAEs) | 48.18 | 31.14 |

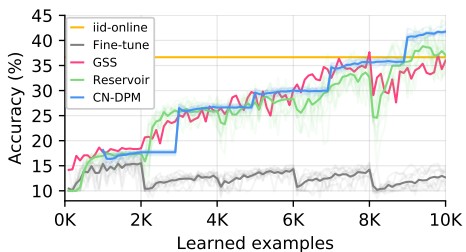

Figure 2: Split-CIFAR10 (0.2 Epoch).

Figure 3: Split-CIFAR100.

**MNIST-SVHN** (Shin et al., 2017). It is a two-stage scenario where the first consists of MNIST, and the second contains SVHN (Netzer et al., 2011). This scenario is different from Split-MNIST; in Split-MNIST, new classes are introduced when transitioning into a new task, whereas the two stages in MNIST-SVHN share the same set of class labels and have different input domains.

**Split-CIFAR10 and Split-CIFAR100**. In Split-CIFAR10, we split CIFAR10 (Krizhevsky & Hinton, 2009) into five tasks in the same manner as Split-MNIST. For Split-CIFAR100, we build 20 tasks, each containing five classes according to the pre-defined superclasses in CIFAR100. The training sets of CIFAR10 and CIFAR100 consist of 50K examples each. Note that most of the previous works (Rebuffi et al., 2017; Zenke et al., 2017; Lopez-Paz & Ranzato, 2017; Aljundi et al., 2019c; Chaudhry et al., 2019a), except Maltoni & Lomonaco (2019), use task information at test time in Split-CIFAR100 experiments. They assign distinct output heads for each task and utilize the task identity to choose the responsible output head at both training and test time. Knowing the right output head, however, the task reduces to 5-way classification. Therefore, our setting is far more difficult than the prior works since the model has to perform 100-way classification only from the given input.

## 4.2 COMPARED METHODS

All the following baselines use the same base network that will be discussed in section 4.3.

**iid-offline and iid-online**. iid-offline shows the maximum performance achieved by combining standard training techniques such as data augmentation, learning rate decay, multiple iterations (up to 100 epochs), and larger batch size. iid-online is the model trained with the same number of epoch and batch size with other CL baselines.

**Fine-tune**. As a popular baseline in the previous works, the base model is naively trained as data enters.

**Reservoir**. As Chaudhry et al. (2019b) show that simple experience replay (ER) can outperform most CL methods, we test the ER with reservoir sampling as a strong baseline. Reservoir sampling randomly chooses a fixed number of samples with a uniform probability from an indefinitely long stream of data, and thus, it is suitable for managing the replay memory in task-free CL. At each training step, the model is trained using a mini-batch from the data stream and another one of the same sizes from the memory.

**Gradient-Based Sample Selection (GSS)**. Aljundi et al. (2019b) propose a sampling method called GSS that diversifies the gradients of the samples in the replay memory. Since it is designed to work in task-free settings, we report the scores in their paper for comparison.

### 4.3 MODEL ARCHITECTURE

**Split-MNIST**. Following Hsu et al. (2018), we use a simple two-hidden-layer MLP classifier with ReLU activation as the base model for classification. The dimension of each layer is 400. For generation experiments, we use VAE, whose encoder and decoder have the same hidden layer configuration with the classifier. Each expert in CN-DPM has a similar classifier and VAE with smaller hidden dimensions. The first expert starts with 64 hidden units per layer and adds 16 units when a new expert is added. For classification, we adjust hyperparameter $\alpha$ such that five experts are created. For generation, we set $\alpha$ to produce 12 experts since more experts produce a better score. We set the memory size in both Reservoir and CN-DPM to 500 for classification and 1000 for generation.

**MNIST-SVHN and Split-CIFAR10/100**. We use ResNet-18 (He et al., 2016) as the base model. In CN-DPM, we use a 10-layer ResNet for the classifier and a CNN-based VAE. The encoder and the decoder of VAE have two CONV layers and two FC layers. We set $\alpha$ such that 2, 5, and 20 experts are created for each scenario. The memory sizes in Reservoir, GSS, and CN-DPM are set to 500 for MNIST-SVHN and 1000 for Split-CIFAR10/100. More details can be found in Appendix C.

### 4.4 RESULTS OF TASK-FREE CONTINUAL LEARNING

All reported numbers in our experiments are the average of 10 runs. Table 1 and 2 show our main experimental results. In every setting, CN-DPM outperforms the baselines by significant margins with reasonable parameter usage. Table 2 and Figure 2 shows the results of Split-CIFAR10 experiments. Since Aljundi et al. (2019b) test GSS using only 10K examples of CIFAR10, which is 1/5 of the whole train set, we follow their setting (denoted by *0.2 Epoch*) for a fair comparison. We also test a Split-CIFAR10 variant where each task is presented for 10 epochs. The accuracy and the training graph of GSS are excerpted from the original paper, where the accuracy is the average of three runs, and the graph is from one of the runs. In Figure 2, the bold line represents the average of 10 runs (except GSS, which is a single run), and the faint lines are the individual runs. Surprisingly, Reservoir even surpasses the accuracy of GSS and proves to be a simple but powerful CL method.

One interesting observation in Table 2 is that the performance of Reservoir degrades as each task is extended up to 10 epochs. This is due to the nature of replay methods; since the same samples are replayed repeatedly as representatives of the previous tasks, the model tends to be overfitted to the replay memory as training continues. This degradation is more severe when the memory size is small, as presented in Appendix I. Our CN-DPM, on the other hand, uses the memory to buffer recent examples temporarily, so there is no such overfitting problem. This is also confirmed by the CN-DPM's accuracy consistently increasing as learning progresses.

In addition, CN-DPM is particularly strong compared to other baselines when the number of tasks increases. For example, Reservoir, which performs reasonably well in other tasks, scores poorly in Split-CIFAR100, which involves 20 tasks and 100 classes. Even with the large replay memory of size 1000, the Reservoir suffers from the shortage of memory (e.g., only 50 slots per task). In contrast, CN-DPM's accuracy is more than double of Reservoir and comparable to that of iid-online.

Table 3 analyzes the accuracy of CN-DPM in Split-CIFAR10/100. We assess the performance and forgetting of individual components. At the end of each task, we measure the test accuracy of the responsible classifier and report the average of such task-wise classifier accuracies as *Classifier (init)*.

We report the average of the task-wise accuracies after learning all tasks as *Classifier (final)*. With little difference between the two scores, we confirm that forgetting barely occurs in the classifiers. In addition, we report the gating accuracy measured after training as *Gating (VAEs)*, which is the accuracy of the task identification performed jointly by the VAEs. The relatively low gating accuracy suggests that CN-DPM has much room for improvement through better density estimates.

Overall, CN-DPM does not suffer from catastrophic forgetting, which is a major problem in regularization and replay methods. As a trade-off, however, choosing the right expert arises as another problem in CN-DPM. Nonetheless, the results show that this new direction is especially promising when the number of tasks is very large.

## 5 CONCLUSION

In this work, we formulated expansion-based task-free CL as learning of a Dirichlet process mixture model with neural experts. We demonstrated that the proposed CN-DPM model achieves great performance in multiple task-free settings, better than the existing methods. We believe there are several interesting research directions beyond this work: (i) improving the accuracy of expert selection, which is the main bottleneck of our method, and (ii) applying our method to different domains such as natural language processing and reinforcement learning.

### ACKNOWLEDGMENTS

We thank Chris Dongjoo Kim and Yookoon Park for helpful discussion and advice. This work was supported by Video Analytics Center of Excellence in AIX center of SK telecom, Institute of Information & communications Technology Planning & Evaluation (IITP) grant (No.2019-0-01082, SW StarLab) and Basic Science Research Program through National Research Foundation of Korea (NRF) funded by the Korea government (MSIT) (2017R1E1A1A01077431). Gunhee Kim is the corresponding author.

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

## A  REVIEW OF DIRICHLET PROCESS MIXTURE MODEL

We review the Dirichlet process mixture (DPM) model and a variational method to approximate the posterior of DPM models in an online setting: Sequential Variational Approximation (SVA) (Lin, 2013).

**Dirichlet Process**. Dirichlet process (DP) is a distribution over distributions that are defined over infinitely many dimensions. DP is parameterized by a concentration parameter $\alpha \in \mathbb{R}^+$ and a base distribution $G_0$. For a distribution $G$ sampled from $\mathrm{DP}(\alpha, G_0)$, the following holds for any finite measurable partition $\{A_1, A_2, ..., A_K\}$ of probability space $\Theta$ (Teh, 2010):

$$(G(A_1), ..., G(A_K)) \sim \mathrm{Dir}(\alpha G_0(A_1), ..., \alpha G_0(A_K)). \tag{9}$$

The stick-breaking process is often used as a more intuitive construction of DP:

$$G = \sum_{k=1}^{\infty} \left( v_k \prod_{l=1}^{k-1} (1 - v_l) \right) \delta_{\phi_k}, v_k \sim \mathrm{Beta}(1, \alpha), \phi_k \sim G_0. \tag{10}$$

Initially, we start with a stick of length one, which represents the total probability. At each step $k$, we cut a proportion $v_k$ off from the remaining stick (probability) and assign it to the atom $\phi_k$ sampled from the base distribution $G_0$. This formulation shows DP is discrete with probability 1 (Teh, 2010). In our problem setting, $G$ is a distribution over expert's parameter space and has positive probability only at the countably many $\phi_k$, which are independently sampled from the base distribution.

**Dirichlet Process Mixture (DPM) Model**. The DPM model is often applied to clustering problems where the number of clusters is not known in advance. The generative process of DPM model is

$$x_n \sim p(\theta_n), \quad \theta_n \sim G, \quad G \sim \mathrm{DP}(\alpha, G_0), \tag{11}$$

where $x_n$ is the $n$-th data, and $\theta_n$ is the $n$-th latent variable sampled from $G$, which itself is a distribution sampled from a Dirichlet process (DP). Since $G$ is discrete with probability 1, the same values can be sampled multiple times for $\theta$. If $\theta_n = \theta_m$, the two data points $x_n$ and $x_m$ belong to the same cluster. An alternative formulation uses the indicator variable $z_n$ that indicates to which cluster the $n$-th data belongs such that $\theta_n = \phi_{z_n}$ where $\phi_k$ is the parameter of $k$-th cluster. The data $x_n$ is sampled from a distribution parameterized by $\theta_n$. For a DP Gaussian mixture model as an example, each $\theta = \{\mu, \sigma^2\}$ parameterizes a Gaussian distribution.

**The Posterior of DPM Models**. The posterior of a DPM model for given $\theta_1, ..., \theta_n$ is also a DP (Teh, 2010):

$$G|\theta_1, ..., \theta_n \sim \mathrm{DP}\left( \alpha + n, \frac{\alpha}{\alpha + n} G_0 + \frac{1}{\alpha + n} \sum_{i=1}^{n} \delta(\theta_i) \right). \tag{12}$$

The base distribution of the posterior, which is a weighted average of $G_0$ and the empirical distribution $\frac{1}{n} \sum_{i=1}^{n} \delta(\theta_i)$, is in fact the predictive distribution of $\theta_{n+1}$ given $\theta_{1:n}$ (Teh, 2010):

$$\theta_{n+1}|\theta_1, ..., \theta_n \sim \frac{\alpha}{\alpha + n} G_0 + \frac{1}{\alpha + n} \sum_{i=1}^{n} \delta(\theta_i). \tag{13}$$

If we additionally condition $x_n$ and reflect the likelihood, we obtain (Neal, 2000):

$$\theta_{n+1}|\theta_1, ..., \theta_n, x_{n+1} \sim \frac{1}{Z} \left( \frac{\alpha}{\alpha + n} \int p(x_{n+1}|\theta) dG_0(\theta) + \frac{1}{\alpha + n} \sum_{i=1}^{n} p(x_{n+1}|\theta_i)\delta(\theta_i) \right), \tag{14}$$

where $Z$ is the normalizing constant. Note that $\theta_{n+1}$ is independent from $x_{1:n}$ given $\theta_{1:n}$.

**Approximation of the Posterior of DPM Models**. Since the exact inference of the posterior of DPM models is infeasible, approximate inference methods are adopted such as Markov chain Monte Carlo (MCMC) (Maceachern, 1994; Escobar & West, 1995; Neal, 2000) or variational inference (Blei & Jordan, 2006; Wang & Dunson, 2011; Lin, 2013). Among many variational methods, the Sequential Variational Approximation (SVA) (Lin, 2013) approximates the posterior as

$$p(G|x_{1:n}) = \sum_{z_{1:n}} p(z_{1:n}|x_{1:n})p(G|x_{1:n}, z_{1:n}) \ \approx \ q(G|\rho, \nu) = \sum_{z_{1:n}} \left( \prod_{i=1}^{n} \rho_{i,z_i} \right) q_{\nu}^{(z)}(G|z_{1:n}), \tag{15}$$

where $p(z_{1:n}|x_{1:n})$ is represented by the product of individual variational probabilities $\rho_{z_i}$ for $z_i$, which greatly simplifies the distribution. Moreover, $p(G|x_{1:n}, z_{1:n})$ is approximated by a stochastic process $q_\nu^{(z)}(G|z_{1:n})$. Sampling from $q_\nu^{(z)}(G|z_{1:n})$ is equivalent to constructing a distribution as

$$\beta_0 D' + \sum_{k=1}^{K} \beta_k \delta_{\phi_k}, \quad D' \sim \text{DP}(\alpha G_0), (\beta_0, \dots, \beta_K) \sim \text{Dir}(\alpha, |C_1^{(z)}|, \dots, |C_K^{(z)}|), \phi_k \sim \nu_k, \quad (16)$$

where $\{C_1^{(z)}, C_2^{(z)}, \dots, C_K^{(z)}\}$ is the partition of $x_{1:n}$ characterized by $z$.

The approximation yields the following tractable predictive distribution:

$$q(\theta'|\rho, \nu) = \mathbb{E}_{q(G|\rho, \nu)}[p(\theta'|G)] = \frac{\alpha}{\alpha + n} G_0(\theta') + \sum_{k=1}^{K} \frac{\sum_{i=1}^{n} \rho_{i,k}}{\alpha + n} \nu_k(\theta'). \quad (17)$$

SVA uses this predictive distribution for sequential approximation of the posterior of $z$ and $\phi$.

$$p(z_{n+1}, \phi^{(n+1)}|x_{1:n+1}) \propto p(x_{n+1}|z_{n+1}, \phi^{(n+1)}) p(z_{n+1}, \phi^{(n+1)}|x_{1:n}) \quad (18)$$

$$\approx p(x_{n+1}|z_{n+1}, \phi^{(n+1)}) q(z_{n+1}, \phi^{(n+1)}|\rho_{1:n}, \nu^{(n)}). \quad (19)$$

While the data is given one by one, SVA sequentially updates the variational parameters; the following $\rho_{n+1}$ and $\nu^{(n+1)}$ at step $n+1$ minimizes the KL divergence between $q(z_{n+1}, \phi^{(n+1)}|\rho_{1:n+1}, \nu^{(n+1)})$ and the posterior:

$$\rho_{n+1,k} \propto \begin{cases} \left(\sum_{i=1}^{n} \rho_{i,k}\right) \int_\theta p(x_{n+1}|\theta) \nu_k^{(n)}(d\theta) & \text{if } 1 \le k \le K \\ \alpha \int_\theta p(x_{n+1}|\theta) G_0(d\theta) & \text{if } k = K+1 \end{cases}, \quad (20)$$

$$\nu_k^{(n+1)}(d\theta) \propto \begin{cases} G_0(d\theta) \prod_{i=1}^{n+1} p(x_i|\theta)^{\rho_{i,k}} & \text{if } 1 \le k \le K \\ G_0(d\theta) p(x_{n+1}|\theta)^{\rho_{n+1,k}} & \text{if } k = K+1 \end{cases}. \quad (21)$$

In practice, SVA adds a new component only when $\rho_{n+1,K+1}$ is greater than a threshold $\epsilon$. It uses stochastic gradient descent to find and maintain the MAP estimation of parameters instead of calculating the whole distribution $\nu_k$:

$$\hat{\phi}_k^{(n+1)} \leftarrow \hat{\phi}_k^{(n)} + \lambda_n(\nabla_{\hat{\phi}_k^{(n)}} \log G_0(\hat{\phi}_k^{(n)}) + \nabla_{\hat{\phi}_k^{(n)}} \log p(x|\hat{\phi}_k^{(n)})), \quad (22)$$

where $\lambda_k^{(n)}$ is a learning rate of component $k$ at step $n$, which decreases as in the Robbins-Monro algorithm.

## B  PRACTICAL ISSUES OF CN-DPM

CN-DPN is designed based on strong theoretical foundations, including the nonparametric Bayesian framework. In this section, we further discuss some practical issues of CN-DPM with intuitive explanations.

**Bounded expansion of CN-DPM**. The number of components in the DPM model is determined by the data distribution and the concentration parameter. If the true distribution consists of $K$ clusters, the number of effective components converges to $K$ under an appropriate concentration parameter $\alpha$. Typically, the number of components is bounded by $O(\alpha \log N)$ (Teh, 2010). Experiments in Appendix H empirically show that CN-DPM does not blindly increase the number of experts.

**The continued increase in model capacity**. Our model capacity keeps increasing as it learns new tasks. However, we believe this is one of the strengths of our method, since it may not make sense to use a fixed-capacity neural network to learn an indefinitely long sequence of tasks. The underlying assumption of using a fixed-capacity model is that the pre-set model capacity is adequate (at least not insufficient) to learn the incoming tasks. On the other hand, CN-DPM approaches the problem in a different direction: *start small and add more as needed*. This property is essential in task-free settings where the total number of tasks is not known. If there are too many tasks than expected, a fixed-capacity model would not be able to learn them successfully. Conversely, if there are fewer

tasks than expected, resources would be wasted. We argue that expansion is a promising direction since it does not need to fix the model capacity beforehand. Moreover, we also introduce an algorithm to prune redundant experts in Appendix D,

**Generality of the concentration parameter**. The concentration parameter controls how sensitive the model is to new data. In other words, it determines *the level of discrepancy* between tasks, that makes the tasks modeled by distinct components. As an example, suppose that we are designing a hand-written alphabet classifier that continually learns in the real world. In the development, we only have the character images for half of the alphabets, i.e., from 'a' to 'm'. If we can find a good concentration parameter $\alpha$ for the data from 'a' to 'm', the same $\alpha$ can work well with novel alphabets (i.e., from 'n' to 'z') because the alphabets would have a similar level of discrepancies between tasks. Therefore, we do not need to access the whole data to determine $\alpha$ if the discrepancy between tasks is steady.

## C  MODEL ARCHITECTURES AND EXPERIMENTAL DETAILS

### C.1  BASE MODELS

#### C.1.1  SPLIT-MNIST

Following Hsu et al. (2018), we use two-hidden-layer MLP classifier with 400 hidden units per layer. For generation tasks, we use a simple VAE with the two-hidden-layer MLP encoder and decoder, where each layer contains 400 units. The dimension of the latent code is set to 32. We use ReLU for all intermediate activation functions.

#### C.1.2  MNIST-SVHN AND SPLIT-CIFAR10/100

We use ResNet-18 (He et al., 2016). The input images are transformed to 32×32 RGB images.

### C.2  CN-DPM

#### C.2.1  SPLIT-MNIST

For the classifiers in experts, we use a smaller version of the base MLP classifier. In the first expert, we set the number of hidden units per layer to 64. In the second or later experts, we introduce 16 new units per layer which are connected to the lower layers of the existing experts. For the encoder and decoder of VAEs, we use a two-layer MLP. The encoder is expanded in the same manner as the classifier. However, we do not share the parameters beyond the encoders; with a latent code of dimension 16, we use the two-hidden-layer MLP decoder as done in the classifier. For generation tasks, we double the size; for example, we set the size of initial and additional hidden units to 128 and 32, respectively.

#### C.2.2  SPLIT-CIFAR10/100

The ResNet-18 base network has eight residual blocks. After passing through 2 residual blocks, the width and height of the feature are halved, and the number of channels is doubled. The initial number of channels is set to 64.

For the classifiers in CN-DPM, we use a smaller version of ResNet that has only four residual blocks and resizes the feature every block. The initial number of channels is set to 20 in the first expert, and four initial channels are added with a new expert. Thus, 4, 8, 16, and 32 channels are added for the four blocks. The first layer of each block is connected to the last layer of the previous block of prior experts.

For the VAEs, we use a simple CNN-based VAEs. The encoder has two 3×3 convolutions followed by two fully connected layers. Each convolution is followed by 2×2 max-pool and ReLU activation. The numbers of channels and hidden units are doubled after each layer. In the first expert, the first convolution outputs 32 channels, while four new channels are added with each new expert.

As done for the VAE in Split-MNIST, each expert's VAE has an unshared decoder with a 64-dimensional latent code. The decoder is the mirrored encoder where $3 \times 3$ convolution is replaced by $4 \times 4$ transposed convolution with a stride of 2.

### C.2.3 MNIST-SVHN

For the classifier, we use ResNet-18 with 32 channels for the first expert and additional 32 channels for each new expert. We use the same VAE as in Split-CIFAR10.

## C.3 EXPERIMENTAL DETAILS

We use the classifier temperature parameter of 0.01 for Split-MNIST, Split-CIFAR10/100, and no temperature parameter on MNIST-SVHN. Weight decay 0.00001 has been used for every model in the paper. Gradients are clipped by value with a threshold of 0.5. All the CN-DPM models are trained by Adam optimizer. During the sleep phase, we train the new expert for multiple epochs with a batch size of 50. In classification tasks, we improve the density estimation of VAEs by sampling 16 latent codes and averaging the ELBOs, following Burda et al. (2015).

### C.3.1 SPLIT-MNIST

The learning rate of 0.0001 and 0.0004 has been used for the classifier and VAE of each expert in the classification task. We use learning rate 0.003 for the VAE of each expert in generation task. In the generation task, we decay the learning rate of the expert by 0.003 before it enters the wake phase. Following the existing works in VAE literature, we use binarized MNIST for the generation experiments. VAEs are trained to maximize Bernoulli log-likelihood in the generation task, while Gaussian log-likelihood is used for the classification task.

### C.3.2 SPLIT-CIFAR10

The learning rate of 0.005 and 0.0002 has been used for the classifier and VAE of each expert in CIFAR10. We decay the learning rate of the expert by 0.1 before it enters the wake phase. VAEs are trained to maximize Gaussian log-likelihood.

### C.3.3 SPLIT-CIFAR100

The learning rate of 0.0002 and 0.0001 has been used for the classifier and VAE of each expert in CIFAR10. We decay the learning rate of the expert by 0.2 before it enters the wake phase. VAEs are trained to maximize Gaussian log-likelihood.

### C.3.4 MNIST-SVHN

The learning rate of 0.0001 and 0.0003 has been used for the classifier and VAE of each expert in CIFAR10. We decay the learning rates of classifier and VAE of each expert by 0.5 and 0.1 before it enters the wake phase. VAEs are trained to maximize Gaussian log-likelihood.

## D PRUNING REDUNDANT EXPERTS

Lin (2013) propose a simple algorithm to prune and merge redundant components in DPM models. Following the basic principle of the algorithm, we provide a pruning algorithm for CN-DPM. First, we need to measure the similarities between experts to choose which expert to prune. We compute the log-likelihood $l_{nk} = p(x_n, y_n | \hat{\phi}_k)$ of each expert $k$ for data $(x_{1:N}, y_{1:N})$. As a result, we can obtain $K$ vectors with $N$ dimensions. We define the similarity $s(k, k')$ between two experts $k$ and $k'$ as the cosine similarity between the two corresponding vectors $l_{\cdot k}$ and $l_{\cdot k'}$, i.e., $s(k, k') = \frac{l_{\cdot k} \cdot l_{\cdot k'}}{|l_{\cdot k}||l_{\cdot k'}|}$. If the similarity is greater than a certain threshold $\epsilon$, we remove one of the experts with smaller $N_k = \sum_n \rho_{n,k}$. The $N_k$ data of the removed expert are added to the remaining experts.

Figure 4 shows an example of an expert pruning. We test CN-DPM on Split-MNIST with an $\alpha$ higher than the optimal value such that more than five experts are created. In this case, seven experts

are created. If we build a similarity matrix as shown in Figure 4b, we can see which pair of experts are similar. We then threshold the matrix at 0.9 in Figure 4c and choose expert pairs (2/3) and (5/6) for pruning. Comparing $N_k$ within each pair, we can finally choose to prune expert 3 and 6. After pruning, the test accuracy marginally drops from 87.07% to 86.01%.

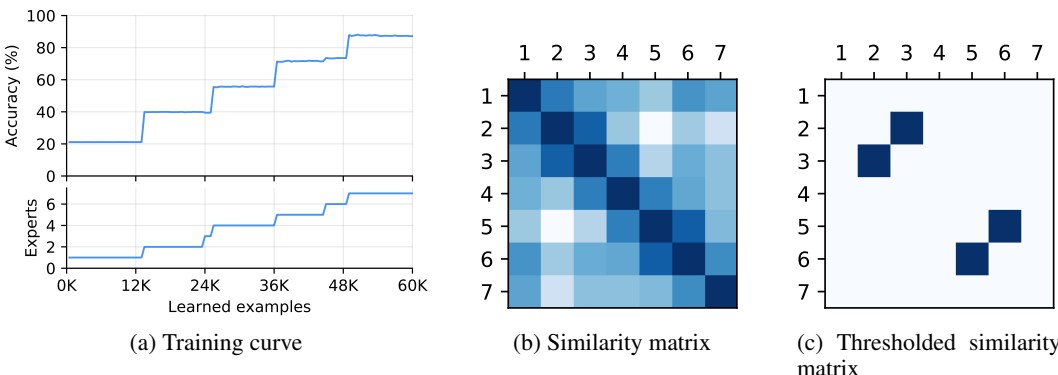

(a) Training curve       (b) Similarity matrix       (c) Thresholded similarity matrix

Figure 4: An example of the expert pruning in the Split-MNIST scenario.

## E   COMPARISON WITH TASK-BASED METHODS ON SPLIT-MNIST

Table 4 compares our method with task-based methods for Split-MNIST classification. All the numbers except for our CN-DPM are excerpted from Hsu et al. (2018), in which all methods are trained for four epochs per task with a batch size of 128. Our method is trained for four epochs per task with a batch size of 10. The model architecture used in compared methods is the same as our baselines: a two-hidden-layer MLP with 400 hidden units per layer. All compared methods use a single output head, and the task information is given at training time but not at test time. For CN-DPM, we test two training settings where the first one uses task information to select experts, while the second one infers the responsible expert by the DPM principle. Task information is not given at test time in both cases.

Notice that regularization methods often suffer from catastrophic forgetting while replay methods yield decent accuracies. Even though the task-free condition is a far more difficult setting, the performance of our method is significantly better than regularization and replay methods that exploit the task description. If task information is available at train time, we can utilize it to improve the performance even more.

Table 4: Comparison with task-based methods on Split-MNIST classification. We report the average of 10 runs with $\pm$ standard error of the mean. The numbers except ours are from Hsu et al. (2018).

| Type | Method | Task labels | Accuracy (%) |
|---|---|---|---|
| Regularization | EWC (Kirkpatrick et al., 2017) | ✓ | $19.80 \pm 0.05$ |
| | Online EWC (Schwarz et al., 2018) | ✓ | $19.77 \pm 0.04$ |
| | SI (Zenke et al., 2017) | ✓ | $19.67 \pm 0.09$ |
| | MAS (Aljundi et al., 2018) | ✓ | $19.52 \pm 0.04$ |
| | LwF (Li & Hoiem, 2017) | ✓ | $24.17 \pm 0.33$ |
| Replay | GEM (Lopez-Paz & Ranzato, 2017) | ✓ | $92.20 \pm 0.12$ |
| | DGR (Shin et al., 2017) | ✓ | $91.24 \pm 0.33$ |
| | RtF (van de Ven & Tolias, 2018) | ✓ | $92.56 \pm 0.21$ |
| Expansion | CN-DPM | ✓ | $93.81 \pm 0.07$ |
| | CN-DPM | ✗ | $93.70 \pm 0.07$ |
| | Upper bound (iid) | | $97.53 \pm 0.30$ |

Table 5: Fuzzy Split-MNIST

| Method | Acc. (%) | Param. |
|---|---|---|
| Fine-tune | $28.41 \pm 0.52$ | 478K |
| Reservoir | $88.64 \pm 0.48$ | 478K |
| CN-DPM | $\mathbf{93.22} \pm 0.07$ | 524K |

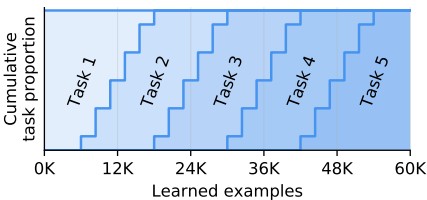

Figure 5: Scenario configuration of Fuzzy Split-MNIST

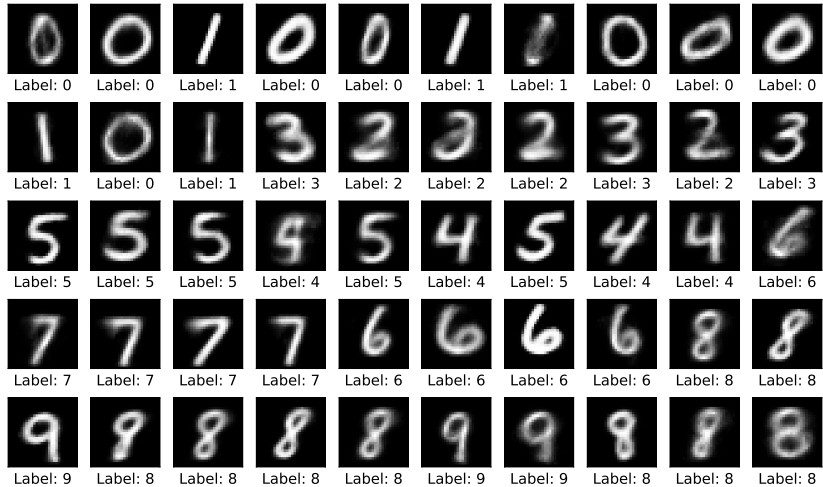

Figure 6: Examples of generation samples by CN-DPM trained on Split-MNIST.

## F    FUZZY SPLIT-MNIST

In addition, we experiment with the case where the task boundaries are not clearly defined, which we call *Fuzzy-Split-MNIST*. Instead of discrete task boundaries, we have transition stages between tasks where the data of existing and new tasks are mixed, but the proportion of the new task linearly increases. This condition adds another level of difficulty since it makes the methods unable to rely on clear task boundaries. The scenario is visualized in Figure 5. As shown in Table 5, CN-DPM can perform continual learning without task boundaries.

## G    GENERATION OF SAMPLES

Even in discriminative tasks where the goal is to model $p(y|x)$, CN-DPM learns the joint distribution $p(x, y)$. Since CN-DPM is a complete generative model, it can generate $(x, y)$ pairs. To generate a sample, we first sample $z$ from $p(z)$ which is modeled by the categorical distribution $\text{Cat}(\frac{N_1}{N}, \frac{N_2}{N}, ..., \frac{N_K}{N})$, i.e., choose an expert. Given $z = k$, we first sample $x$ from the generator $p(x; \phi_k^G)$, and then sample $y$ from the discriminator $p(y|x; \phi_k^D)$. Figure 6 presents 50 sample examples generated from a CN-DPM trained on Split-MNIST for a single epoch. We observe that CN-DPM successfully generates examples of all tasks with no catastrophic forgetting.

## H    EXPERIMENTS WITH LONGER CONTINUAL LEARNING SCENARIOS

We present experiments with much longer continual learning scenarios on Split-MNIST, Split-CIFAR10 and Split-CIFAR100 in Table 6, 7 and 8, respectively. We report the average of 10 runs with $\pm$ standard error of the mean. To compare with the default 1-epoch scenario, we carry out experiments that repeat each task 10 times, which are denoted *10 Epochs*. In addition, we also present

the results of repeating the whole scenario 10 times, which are denoted *1 Epoch ×10*. For example, in Split-MNIST, the *10 Epochs* scenario consists of 10-epoch 0/1, 10-epoch 2/3, ..., 10-epoch 8/9 tasks. On the other hand, the *1 Epoch ×10* scenario revisits each task multiple times, i.e., 1-epoch 0/1, 1-epoch 2/3, ..., 1-epoch 8/9, 1-epoch 0/1, ..., 1-epoch 8/9. We use the same hyperparameters tuned for the 1-epoch scenario.

We find that the accuracy of Reservoir drops as the length of each task increases. As mentioned in the main text, this phenomenon seems to be caused by overfitting on the samples in the replay memory. Since only a small number of examples in the memory represent each task, replaying them for a long period degrades the performance. On the other hand, the performance of our CN-DPM improves as the learning process is extended.

In the *1 Epoch ×10* setting, CN-DPM shows similar performance with *10 Epoch* since the model sees each data point 10 times in both scenarios. On the other hand, Reservoir's scores in the *1 Epoch ×10* largely increase compared to both *1 Epoch* and *10 Epoch* This difference can be explained by how the replay memory changes while training progresses. In the *10 Epoch* setting, if a task is finished, it is not visited again. Therefore, the examples of the task in the replay memory monotonically decreases, and the remaining examples are replayed repeatedly. As the training progresses, the model is overfitted to the old examples in the memory and fails to generalize in the old tasks. In contrast, in *1 Epoch ×10* setting, each task is revisited multiple times, and each time a task is revisited, the replay memory is also updated with the new examples of the task. Therefore, the overfitting problem in the old tasks is greatly relieved.

Another important remark is that CN-DPM does not blindly increase the number of experts. If we add a new expert at every constant steps, we would have 10 times more experts in the longer scenarios. However, this is not the case. CN-DPM determines whether it needs a new expert on a data-by-data basis such that the number of experts is determined by the task distribution, not by the length of training.

Table 6: Experiments with longer training episodes on Split-MNIST

| Method | 1 Epoch | | 10 Epochs | | 1 Epoch ×10 | |
|---|---|---|---|---|---|---|
| | Acc. (%) | Param. | Acc. (%) | Param. | Acc. (%) | Param. |
| iid-offline | $98.63 \pm 0.01$ | 478K | $98.63 \pm 0.01$ | 478K | $98.63 \pm 0.01$ | 478K |
| iid-online | $96.18 \pm 0.19$ | 478K | $97.67 \pm 0.05$ | 478K | $97.67 \pm 0.05$ | 478K |
| Fine-tune | $19.43 \pm 0.02$ | 478K | $19.68 \pm 0.01$ | 478K | $20.27 \pm 0.26$ | 478K |
| Reservoir | $85.69 \pm 0.48$ | 478K | $78.82 \pm 0.71$ | 478K | $92.06 \pm 0.11$ | 478K |
| CN-DPM | $\mathbf{93.23} \pm 0.09$ | 524K | $\mathbf{94.39} \pm 0.04$ | 524K | $\mathbf{94.15} \pm 0.04$ | 616K |

Table 7: Experiments with longer training episodes on Split-CIFAR10

| Method | 1 Epoch | | 10 Epochs | | 1 Epoch ×10 | |
|---|---|---|---|---|---|---|
| | Acc. (%) | Param. | Acc. (%) | Param. | Acc. (%) | Param. |
| iid-offline | $93.17 \pm 0.03$ | 11.2M | $93.17 \pm 0.03$ | 11.2M | $93.17 \pm 0.03$ | 11.2M |
| iid-online | $62.79 \pm 1.30$ | 11.2M | $83.19 \pm 0.27$ | 11.2M | $83.19 \pm 0.27$ | 11.2M |
| Fine-tune | $18.08 \pm 0.13$ | 11.2M | $19.31 \pm 0.03$ | 11.2M | $19.33 \pm 0.03$ | 11.2M |
| Reservoir | $44.00 \pm 0.92$ | 11.2M | $43.82 \pm 0.53$ | 11.2M | $\mathbf{51.44} \pm 0.42$ | 11.2M |
| CN-DPM | $\mathbf{45.21} \pm 0.18$ | 4.60M | $\mathbf{46.98} \pm 0.18$ | 4.60M | $47.10 \pm 0.16$ | 4.60M |

## I  EXPERIMENTS WITH DIFFERENT MEMORY SIZES

In Split-CIFAR10/100 experiments in the main text, we set the memory size of Reservoir and CN-DPM to 1000, following Aljundi et al. (2019b). Table 9 compares the experimental results with different memory sizes of 500 and 1000 on Split-CIFAR10/100. Compared to Reservoir, whose performance drops significantly with smaller memory, CN-DPM's accuracy drop is relatively marginal.

Table 8: Experiments with longer training episodes on Split-CIFAR100

| Method | 1 Epoch | | 10 Epochs | | 1 Epoch $\times 10$ | |
|---|---|---|---|---|---|---|
| | Acc. (%) | Param. | Acc. (%) | Param. | Acc. (%) | Param. |
| iid-offline | $73.80 \pm 0.11$ | 11.2M | $73.80 \pm 0.11$ | 11.2M | $73.80 \pm 0.11$ | 11.2M |
| iid-online | $20.46 \pm 0.30$ | 11.2M | $54.58 \pm 0.27$ | 11.2M | $54.58 \pm 0.27$ | 11.2M |
| Fine-tune | $2.43 \pm 0.05$ | 11.2M | $3.99 \pm 0.03$ | 11.2M | $4.30 \pm 0.02$ | 11.2M |
| Reservoir | $10.01 \pm 0.35$ | 11.2M | $6.61 \pm 0.20$ | 11.2M | $14.53 \pm 0.35$ | 11.2M |
| CN-DPM | $\mathbf{20.10} \pm 0.12$ | 19.2M | $\mathbf{20.95} \pm 0.09$ | 19.2M | $\mathbf{20.67} \pm 0.13$ | 19.2M |

Table 9: Experiments with different memory sizes.

| Method | Memory | Split-CIFAR10 Acc. (%) | | Split-CIFAR100 Acc. (%) | |
|---|---|---|---|---|---|
| | | 1 Epoch | 10 Epoch | 1 Epoch | 10 Epoch |
| Reservoir | 500 | $33.53 \pm 1.03$ | $34.46 \pm 0.49$ | $6.24 \pm 0.25$ | $4.99 \pm 0.09$ |
| CN-DPM | 500 | $\mathbf{43.07} \pm 0.16$ | $\mathbf{47.01} \pm 0.22$ | $\mathbf{19.17} \pm 0.13$ | $\mathbf{20.77} \pm 0.11$ |
| Reservoir | 1000 | $44.00 \pm 0.92$ | $43.82 \pm 0.53$ | $10.01 \pm 0.35$ | $6.61 \pm 0.20$ |
| CN-DPM | 1000 | $\mathbf{45.21} \pm 0.18$ | $\mathbf{46.98} \pm 0.18$ | $\mathbf{20.10} \pm 0.12$ | $\mathbf{20.95} \pm 0.09$ |

## J  THE EFFECT OF CONCENTRATION PARAMETER

Table 10 shows the results of CN-DPM on Split-MNIST classification according to the concentration parameter $\alpha$, which defines the prior of how sensitive CN-DPM is to new data. With a higher $\alpha$, an expert tends to be created more easily. In the experiment reported in the prior sections, we set $\log \alpha = -400$. At $\log \alpha = -600$, too few experts are created, and the accuracy is rather low. As $\alpha$ increases, the number of experts grows along with the accuracy. Although the CN-DPM model is task-free and automatically decides the task assignments to experts, we still need to tune the concentration parameter to find the best balance point between performance and model capacity, as all Bayesian nonparametric models require.

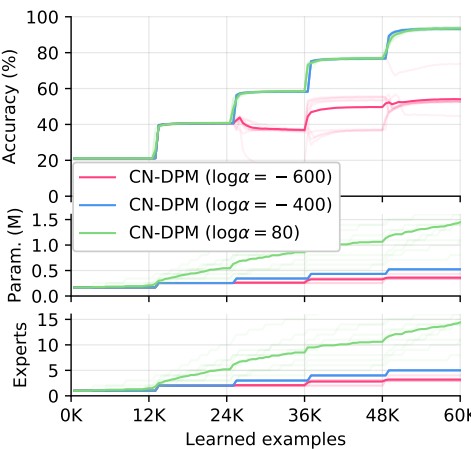

Table 10: The effects of concentration parameter $\alpha$.

| $\log \alpha$ | Acc. (%) | Experts | Param. |
|---|---|---|---|
| $-600$ | $54.04 \pm 2.22$ | $3.20 \pm 0.13$ | 362K |
| $-400$ | $93.23 \pm 0.09$ | $5.00 \pm 0.00$ | 524K |
| $80$ | $93.54 \pm 0.21$ | $14.4 \pm 1.35$ | 1.44M |

Figure 7: The effects of concentration parameter $\alpha$.

## K  THE EFFECT OF PARAMETER SHARING

Table 11 compares when the parameters are shared between experts and when they are not shared. By sharing the parameters, we could reduce the number of parameters by approximately 38% without sacrificing accuracy.

Table 11: The effects of parameter sharing.

| Model | Acc. (%) | Experts | Param. |
|---|---|---|---|
| CN-DPM | $93.23 \pm 0.09$ | 5 | 524K |
| CN-DPM w/o PS | $93.30 \pm 0.24$ | 5 | 839K |

## L   TRAINING GRAPHS

Figure 8 shows the training graphs of our experiments. In addition to the performance metrics, we present the number of experts in CN-DPM and compare the total number of parameters with the baselines. The bold lines represent the average of the 10 runs while the faint lines represent individual runs.

Figure 9 and Figure 10 show how the accuracy of each task changes during training. We also present the average accuracy of learned tasks at the bottom right.

## M   COMPARISON WITH THE CURL

Continual Unsupervised Representation Learning (CURL) (Rao et al., 2019) is a parallel work that shares some characteristics with our CN-DPM in terms of model expansion and short-term memory. However, there are several key differences that distinguish our method from CURL, which will be elaborated in this section. Following the notations of Rao et al. (2019), here $y$ denotes the cluster assignment, and $z$ denotes the latent variable.

**1. The Generative Process**. The primary goal of CURL is to continually learn a unified latent representation $z$, which is shared across all tasks. Therefore, the generative model of CURL explicitly consists of the latent variable $z$ as summarized as follows:

$$p(x, y, z) = p(y)p(z|y)p(x|z) \text{ where } y \sim \text{Cat}(\pi), \ z \sim \mathcal{N}(\mu_z(y), \sigma_z^2(y)), \ x \sim \text{Bernoulli}(\mu_x(z)).$$

The overall distribution of $z$ is the mixture of Gaussians, and $z$ includes the information of $y$ such that $x$ and $y$ are conditionally independent given $z$. Then, $z$ is fed into a single decoder network $\mu_x$ to generate the mean of $x$, which is modeled by a Bernoulli distribution. On the other hand, the generative version of CN-DPM, which does not include classifiers, has a simpler generative process:

$$p(x, y) = p(y)p(x|y) \text{ where } y \sim \text{Cat}(\pi), \ x \sim p(x|y).$$

The choice of $p(x|y)$ here is not necessarily restricted to VAEs (Kingma & Welling, 2014); one may use other kinds of explicit density models such as PixelRNN (Oord et al., 2016). Even if we use VAEs to model $p(x|y)$, the generative process is different from CURL:

$$p(x, y, z) = p(y)p(z)p(x|y, z) \text{ where } y \sim \text{Cat}(\pi), \ z \sim \mathcal{N}(0, I), \ x \sim \text{Bernoulli}(\mu_x^y(z)).$$

Unlike CURL, CN-DPM generates $y$ and $z$ independently and maintains a separate decoder $\mu_x^y$ for each cluster $y$.

**2. The Necessity for Generative Replay in CURL**. CURL periodically saves a copy of its parameters and use it to generate samples of learned distribution. The generated samples are played together with new data such that the main model does not forget previously learned knowledge. This process is called generative replay. The generative replay is an essential element in CURL, unlike our CN-DPM. CURL assumes a factorized variational posterior $q(y, z|x) = q(y|x)q(z|x, y)$ where $q(y|x)$ and $q(z|x, y)$ are modeled by separate output heads of the encoder neural network. However, the output head for $q(y|x)$ is basically a gating network that could be vulnerable to catastrophic forgetting, as mentioned in Section 3.1. Moreover, CURL shares a single decoder $\mu_x$ across all tasks. As a consequence, expansion alone is not enough to stop catastrophic forgetting, and CURL needs another CL method to prevent catastrophic forgetting in the shared components. This is the main reason why the generative replay is crucial in CURL. As shown in the ablation test of Rao et al. (2019), the performance of CURL drops without the generative replay. In contrast, the components of CN-DPM are separated for each task (although they may share low-level representations) such that no additional treatment is needed.

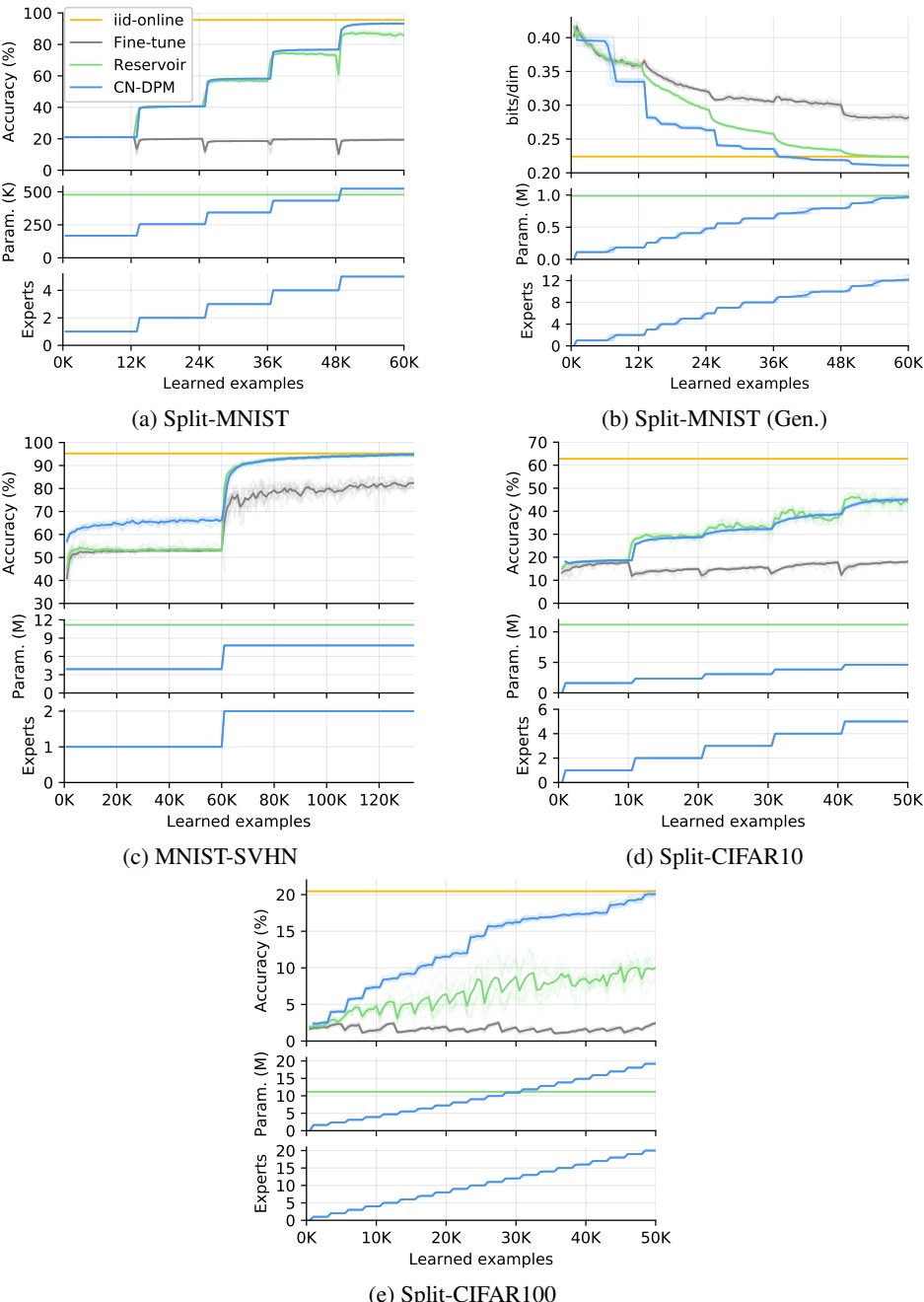

Figure 8: Full training graphs.

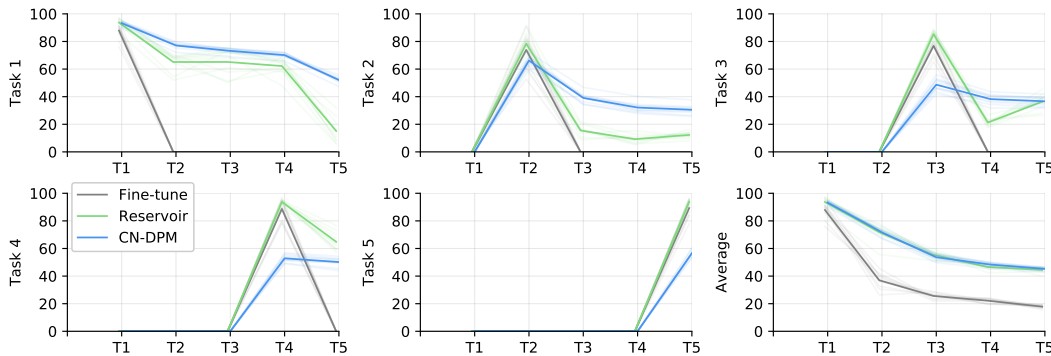

Figure 9: Accuracy for each task in Split-CIFAR10.

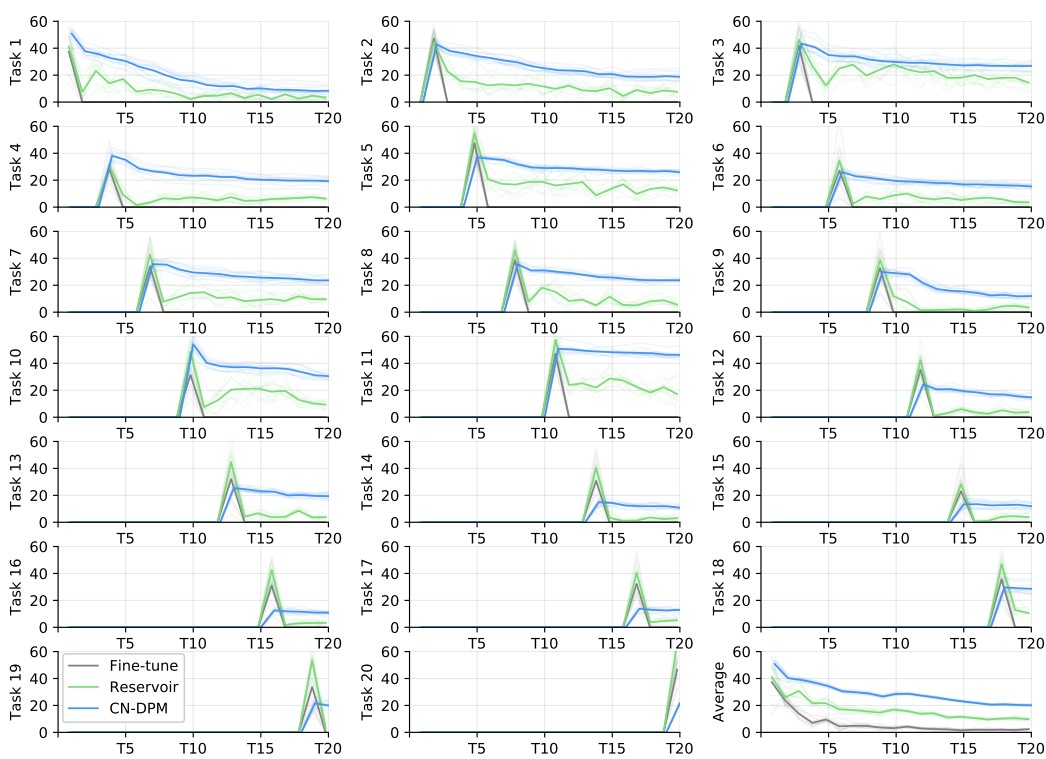

Figure 10: Accuracy for each task in Split-CIFAR100.

