# OpenReview forum: "A Neural Dirichlet Process Mixture Model for Task-Free Continual Learning"
_ICLR.cc/2020/Conference — Accept (Poster)_

### Official Review · AnonReviewer2 · 2019-10-22
**Official Blind Review #2**

**Rating:** 6

**Review:**

This paper proposes Continual Neural Dirichlet Process Mixture Model (CN-DPM) to solve task-free continual learning. The core idea is to employ Dirichlet process mixture model to create novel experts in online fashion when task distributions change.  The proposed method is validated on various tasks and demonstrated to perform well compared to the other baselines.

Overall I find this paper to be well-written and the experiments are conducted thoroughly. The method is compared to proper baselines in various settings, and the paper describes detailed experimental settings and architectural choices to help readers willing to reproduce. I’ve gone through the appendix and they provide enough additional experiments to support the author’s claim.

The main algorithm itself cannot be considered to be novel. DPM or other Bayesian nonparametric models have been extensively used for the problems requiring to adapt the model size according to the change of data. Nevertheless, the application of DPM in task-free continual learning context seems to be considered as a contribution.

I have much experience in implementing Bayesian nonparametric models with parametric distributions and compared various methods to conduct the posterior inference of them. In my experience, even for the low-dimensional parametric models, the posterior inference algorithms for DPM usually suffer from local optima, and the sequential methods such as SVA depends heavily on the data processing order. According to the experimental setting presented in the paper, the algorithm goes through a single pass over the data stream, yet still able to reasonably train (deep) neural networks and identify mixture components jointly. Do you have any intuition about how this becomes feasible?

I don’t fully understand why generative modeling is required. In page 4 the authors stated that the generative model prevents catastrophic forgetting. But in my understanding, using expert-specific parameters is the part that prevents catastrophic forgetting, not the generative model itself. Learning generative model in online fashion may work well in simple structured data such as MNIST,  but I highly doubt that the generative model could be trained properly for CIFAR10 or CIFAR100, especially in online setting. My concern is that learning generative model part may even impede the discriminative learning. Could you elaborate more on this?

Another minor concern is the way the concentration parameter alpha is selected. The authors stated that they chose proper value of alpha according to the number of tasks known in advance. I think this does not make sense. Alpha should also be inferred along with other parameters, or fixed to non-informative value if the performance of the algorithm is not very sensitive to the choice of alpha.

I think it would be more helpful to show how the task-assignment p(z=k|x) is learned. For instance, the clustering accuracy according to p(z=k|x) against the ground-truth task label can be measured, or at least qualitatively show what examples were assigned to each task.

**Experience Assessment:**

I have published in this field for several years.

**Review Assessment: Checking Correctness Of Derivations And Theory:**

N/A

**Review Assessment: Checking Correctness Of Experiments:**

I assessed the sensibility of the experiments.

**Review Assessment: Thoroughness In Paper Reading:**

I read the paper at least twice and used my best judgement in assessing the paper.

---

> ### Author Response · Authors · 2019-11-13
> **Response to Reviewer 2**
>
> We appreciate the thoughtful review. Below, we present our answers to the questions.
>
>
> 1. Feasibility of the posterior inference
>
> This is closely related to the first question of R1. CN-DPM may not be successful when the discrepancy between tasks is too marginal. Conversely, the posterior inference gets easier as the discrepancy between tasks becomes larger.
>
> We believe that most existing continual learning benchmarks are defined to have enough discrepancies between the tasks. In the class-incremental scenarios (e.g. Split-CIFAR100), the difference between the labels largely separates the tasks. In the domain-shifting scenario (e.g. MNIST-SVHN), the obvious differences between grayscale and color images distinguish the tasks.
>
>
> 2. The necessity of generative models
>
> In task-free continual learning, it becomes a critical problem at test time to choose the right expert for a given data point since no task information is available. For example, if M experts are created during training, choosing an expert for a test data is a type of M-class classification problem. If we use a single gate network to solve this problem, catastrophic-forgetting may occur in the gating network. On the other hand, we propose the combination of generative models as a forgetting-free gating mechanism since each generative model is expert-specific.
>
>
> 3. Concerns on learning generative models
>
> We agree that the generative models would not be trained properly in a fully online scenario. However, to be precise, each expert in CN-DPM is not trained in a fully online manner; during the sleep phase, each expert is trained on data in the STM (with a size of 500 or 1000) for multiple iterations. Despite the small size of STM, we empirically confirmed that the sleep phase provides enough bootstrapping for the generative models to be functional.
>
>
> 4. About the known concentration parameter
>
> As the review mentioned, it is possible to infer the value of alpha, although it could be challenging in an online setting like ours. Instead, we want to emphasize that knowing the value of alpha ahead of time is different from knowing the number of tasks beforehand. It is rather knowing the degree of discrepancy between tasks. As long as the discrepancies between tasks remain constant, the model can still learn indefinitely many tasks. Also note that the last paragraph of Appendix B explains the underlying assumption behind choosing the concentration parameter with a concrete example.
>
>
> 5. More explanation on learning p(z=k|x)
>
> CN-DPM does not learn $p(z=k|x)$ directly. Instead, it has a set of generative models that independently learns $p(x|z=k)$. For a given task, only one generative model of the responsible expert is trained. Then, $p(z=k|x)$ is simply computed as $\frac{p(x|z=k) p(z=k)}{\sum_{k’} p(x|z=k’) p(z=k’)}$.
>
> We also reported the gating accuracy of learned $p(z=k|x)$ on CIFAR10/100 in Table 3 as “VAE” (which is updated to “Gating (VAEs)”). The gating accuracy is computed against the ground-truth task label after the whole scenario has finished.

---

### Official Review · AnonReviewer3 · 2019-10-22
**Official Blind Review #3**

**Rating:** 6

**Review:**

Summary: The paper proposes to use a Bayesian nonparametric mixture model for task-free (without explicit task labels) continual learning. The main idea is to use an expansion-based model where the number of mixture components (experts) adapts to the training data/tasks. Specifically, a Dirichlet Process Mixture Model (DPMM) consisting of a set of neural network experts is used. Empirical results demonstrate improved performance on three different datasets over some of the baselines.

I find the methodological contribution in the paper to be somewhat limited since the main idea of the model was initially proposed in the prior work (cited in the main paper): Dahual Lin - “Online Learning of Nonparametric Mixture Models via SVA”. In fact, the paper claims its contribution is expansion-based task-free continual learning. However, this “task-free characteristic” is the contribution of SVA based inference. Nevertheless, I do like how the existing SVA based inference has been adapted from an online learning setting to a more general continual learning setting by using various approximations/tricks (like short-term memory with wake-sleep training, point estimates).

Pros:
- The idea of using a nonparametric model for CL is interesting and can lead to follow-up work.
- Results show that the approach works well.
- The code has been released.

Overall I am inclining towards voting for acceptance if the authors could address my following questions:

- Could you comment on the creation of test data? It is not clear to me how the model is evaluated if the task-boundaries are not known a priori. Shouldn’t the evaluation be based on tasks? How are you evaluating catastrophic forgetting? I am interested to know what was the test accuracy for the task for which the training data was seen early on during the training.

- It seems to me that the method works on the assumption that the number of data points for each task is at least M (size of STM) and moreover, that these data points appear together sequentially. The method should be sensitive to the size of the STM. How are you choosing M? Would the framework work if data points for each task do not appear together?

- Assuming you have clear task boundaries, how would you adapt this framework? Was the model compared to other methods that assume known task-boundaries like (VCL, EWC, Memory Replay)?

Other comments:

- The method is inspired by a Bayesian framework but calling it Bayesian wouldn’t be fair since only a point estimate is being learned for parameters. This is important to distinguish since there are other methods that are fully Bayesian like Nguyen et. al. “Variational Continual Learning”  (although such methods may have other pros and cons)

- The samples from the base distribution of the posterior (v) are not iid anymore due to lateral connections b/w the representations. Do you think the theoretical result in Appendix B that the number of clusters is upper bounded by O(alpha*logN) is still valid?


**Experience Assessment:**

I have published one or two papers in this area.

**Review Assessment: Checking Correctness Of Derivations And Theory:**

I carefully checked the derivations and theory.

**Review Assessment: Checking Correctness Of Experiments:**

I assessed the sensibility of the experiments.

**Review Assessment: Thoroughness In Paper Reading:**

I read the paper thoroughly.

---

> ### Author Response · Authors · 2019-11-13
> **Response to Reviewer 3**
>
> We appreciate your positive and thoughtful review. Please check appendix E and L of the updated draft for added experiment and graphs. Below are our answers to your questions.
>
>
> 1. Evaluation on each task
>
> [1] and [2] evaluate their models only on the tasks that have been presented so far. On the other hand, we always evaluate our model on the entire test set following [3] (no matter whether presented or not). Specifically, the learning curves of Figure 2 show how the accuracy on the entire CIFAR10 test set increases as more tasks are learned.
>
> Table 3 shows that forgetting is minimal in the classifiers of CN-DPM. At the end of each task, we measure the test accuracy of the responsible classiﬁer (labeled as “Classiﬁer (init)”). We also measure the accuracies after learning all tasks (labeled as “Classiﬁer (ﬁnal)”). We observe little difference between the two scores, which conﬁrms that forgetting barely occurs in the classiﬁers.
>
> In appendix L of the updated manuscript, we add more learning curves following [1] and [2]. For Split-CIFAR10/100, we report (i) how the test accuracy of each task changes during training and (ii) how the average test accuracy of learned tasks decreases. Although each expert is hardly updated after their assigned task, the accuracies gradually decrease since the gating performance degrades as an increasing number of VAEs are involved.
>
>
> 2. The STM size
>
> There is a tradeoff in the STM size: as the STM size grows, the performance generally increases, but we need a stronger assumption on the data stream. In our paper, we simply set the STM size to 500 for MNIST and 1000 for the others, following the size of replay memory in [3]. We believe this is a reasonable setting since it is about 1-2% of the training set size.
>
> Appendix I presents the results of Split-CIFAR10/100 with a smaller STM of size 500. Compared to Reservoir whose performance drops significantly, CN-DPM keeps competitive scores even with a halved STM.
>
> Our basic assumption on tasks is that the data points for each task appear together. Otherwise, we may need to allow each task being revisited multiple times, for which we performed experiments in appendix H.
>
>
> 3. CN-DPM with known task boundaries
>
> Originally, Appendix E compared the performance of CN-DPM with other task-based regularization/replay methods (EWC, Online EWC, SI, MAS, LwF, GEM, DGR, and RtF) on the Split-MNIST scenario. The compared methods are applied to a single-headed neural network such that task information is not needed at test time. Despite this handicapped setting, CN-DPM outperformed the task-based methods by significant margins thanks to the expansion paradigm.
>
> In the updated draft, we add an additional experiment for CN-DPM with known task boundaries to appendix E. Given the task boundaries, we do not need to infer the responsible expert. Instead, we simply expand the model at every task boundary and train a new expert for the new task. With the task boundary, the average accuracy further increases from 93.70 to 93.81.
>
>
> 4. The term “Bayesian”
>
> We admit that our method is not “fully Bayesian” as in VCL. Given that our method uses the DPM as an overall framework and performs MAP estimation for experts, we believe that a “Bayesian framework with MAP estimation” may be the best term describing our method. We will clarify the distinction more strictly.
>
>
> 5. The effect of lateral connections
>
> It is a legitimate point that parameter sharing makes the posterior of φ no longer independent anymore. However, the argument of [4] that the number of clusters is “typically” upper bounded by $O(\alpha \log N)$ is a trend, rather than a theoretical result. More precisely, the expected number of clusters in the Chinese restaurant process, which is the prior of DPM, is $\alpha \log N$. In other words, $\alpha \log N$ is the expected number of clusters before we see any data. Therefore, depending on the distribution of the data, the number of clusters can be larger than $O(\alpha * \log N)$.
>
>
> [1] Arslan Chaudhry, Marc’Aurelio Ranzato, Marcus Rohrbach, and Mohamed Elhoseiny. Efficient Lifelong Learning with A-GEM. ICLR, 2019.
> [2] David Lopez-Paz and Marc’Aurelio Ranzato. Gradient Episodic Memory for Continual Learning. NeurIPS 2017.
> [3] Rahaf Aljundi, Min Lin, Baptiste Goujaud, and Yoshua Bengio. Gradient based sample selection for online continual learning. arXiv, (1903.08671v4), 2019.
> [4] Yee Whye Teh. Dirichlet process. Springer, Encyclopedia of Machine Learning:280–287, 2010.

---

### Official Review · AnonReviewer1 · 2019-10-27
**Official Blind Review #1**

**Rating:** 8

**Review:**

The paper proposes an elegant method for task-free continual learning problems. It has nicely pointed out that the conventional continual learning algorithms had the limitation of knowing the task boundaries. Followings are my summary.

Summary:
By applying DPM, the authors proposed a method of automatically determining whether to add a new expert for a new task or train the existing experts. While the Dirichlet Process Mixture (DPM) is not new, applying such nonparametric method to continual learning is new. The experimental results are impressive given the single-epoch setting.

Pros:
1. Good experimental results for the task-free setting, in which no information about task boundaries is given. Particularly, even with smaller memory-usage than the experience replay (ER) methods, the proposed method achieves better results.
2. Many past work should suffer from increased number of tasks due to the model capacity limit, but the proposed method efficiently expands the model capacity.
3. Writing flow is good and is easy to follow.

Cons & Questions:
1. I am not sure whether the proposed methods should work well for "all" cases. Is there any cases in which the proposed DPM would fail?
2. What happens when you actually know the task boundaries? Would following the framework with known number of experts also excel other methods?
3. Can you apply this to the RL setting?


**Experience Assessment:**

I have published one or two papers in this area.

**Review Assessment: Checking Correctness Of Derivations And Theory:**

I assessed the sensibility of the derivations and theory.

**Review Assessment: Checking Correctness Of Experiments:**

I assessed the sensibility of the experiments.

**Review Assessment: Thoroughness In Paper Reading:**

I made a quick assessment of this paper.

---

> ### Author Response · Authors · 2019-11-13
> **Response to Reviewer 1**
>
> Thank you for the constructive and encouraging comments. We have updated appendix E with an additional experiment and provide our responses below.
>
>
> 1. Any cases in which the proposed DPM would fail
>
> CN-DPM may not be successful when the discrepancy between tasks is too marginal. Since the model cannot access previous tasks in continual learning scenarios, new data need to be sufficiently different such that it is classified as new and sent to the STM. Otherwise, this slightly different data is sent to an existing expert causing weak forgetting.
>
>
> 2. Known task boundaries
>
> Originally, Appendix E compared the performance of CN-DPM with other task-based regularization/replay methods (EWC, Online EWC, SI, MAS, LwF, GEM, DGR, and RtF) on the Split-MNIST scenario. The compared methods are applied to a single-headed neural network such that task information is not needed at test time. Despite this handicapped setting, CN-DPM outperformed the task-based methods by significant margins thanks to the expansion paradigm.
>
> In the updated draft, we add an additional experiment for CN-DPM with known task boundaries to appendix E. Given task boundaries, we do not need to infer the responsible expert. Instead, we simply expand the model at every task boundary and train a new expert for the new task. With the task boundary, the average accuracy further increases from 93.70 to 93.81.
>
>
> 3. Applications to the RL setting
>
> We believe that our method has great potential to be applied to RL settings since it is a general framework to learn distributions incrementally. As an example, CN-DPM may be applied to policy gradient methods; each expert consists of a policy network, which predicts an action for a given state, and a generative model, which recognizes whether the expert is responsible for the given state.

---

### Public Comment · ~Erin_Grant1 · 2019-11-06
**The claim that "task-free continual learning" is a novel contribution neglects a prior work [1] which, while cited, is misrepresented in this manuscript.**

We congratulate the authors on their submission.

However, we would like to note that the claim that "expansion-based, task-free continual learning" is a novel contribution ignores a prior work [1] that has been accepted to NeurIPS 2019. The authors cite [1] briefly in the related work section, stating that [1] does not extend beyond meta-learning without going into significant detail. However, in [1], we also adapt the sequential variational approximation approach of Lin et al. (2013) for the Dirichlet process mixture (DPM) and present derivations mirroring those in the submission under review.

In the related work section, the submission claims that [1] "...lacks a generative component which is a crucial element to complete the DPM formulation." While we demonstrate parameter estimation in a mixture of regularized convnet classifiers (as opposed to using a generative component), the procedure remains a "sound" definition of a DPM model. Our approach is furthermore not inconsistent with a mixture of Bayes classifiers (i.e., including a generative component p(x | φ) as in the submission), as it relies only on gradient-based optimization of the components (though we do not test such an architecture empirically). More importantly, [1] also addresses the catastrophic forgetting problem with a DPM-based expansion approach.

The main difference lies in the fact that we apply the algorithm in [1] to the few-shot learning setting, in which the learner receives a batch of data that is known to come from the same domain, and some amount of domain-specific fine-tuning is allowed (i.e., for the bilevel optimization approach of meta-learning). This allows the computation of the assignments to experts (ρ in the submission; γ in [1]) and the update for expert parameters (line 22 in Algo. 1 in the submission; M-STEP in [1]) to be computed using separate data points. The present submission, in contrast, computes these quantities with the same data point; it further does not allow domain-specific fine-tuning. As such, [1] can be immediately applied to the setting of this paper (with the special case of setting the number of fine-tuning steps to 0, and computing the responsibilities γ and the M-STEP using the same datapoint).

In summary, we believe the methodological differences are not as great as claimed in the current submission, and that most of the differences lie in the experimental setup.

Regards,
Authors of [1]

[1] Jerfel, G., Grant, E., Griffiths, T. L., and Heller, K. A. (2019). Reconciling meta-learning and continual learning with online mixtures of tasks. In NeurIPS 2019. https://arxiv.org/abs/1812.06080

---

> ### Author Response · Authors · 2019-11-08
> **Clarification of our novelty over [1].**
>
> We appreciate the authors of [1] for the detailed comments. We acknowledge that the differentiation with [1] was rather cursory in the initial submission due to the page limitation. This will be clarified in the final draft.
>
> First of all, we were surprised to find that our work has technical similarity to [1] (i.e. the use of DPM) as a parallel submission to NeurIPS 2019 (unfortunately, this work was not accepted even with very positive reviews mainly due to experimental evaluation, which was substantially improved in this ICLR submission).
>
> However, we believe there are three significant differences between our work and [1].
>
> 1. The form of continual learning and the way DPM is applied are different.
> In our formulation, a data point to be fed into DPM is a single example (i.e. an image), whereas a datapoint in [1] corresponds to a task (i.e. a set of support and query examples). Thus, at both training and test time, the model of [1] receives data on a task-by-task basis and computes the responsibility for each task, not for each data sample as in our CN-DPM. That is, task-agnostic continual learning takes place at the meta-level in [1]: similar tasks are grouped into a “super-task” for which a meta-expert is responsible, and the super-tasks are automatically discovered by the DPM framework. Therefore, [1] is a method for continual "few-shot" learning rather than general continual learning. Moreover, it cannot be generalized into a standard continual learning setting for the reason described in the following paragraph.
>
> 2. Our generative gating mechanism, which lacks in [1], is critical for the test-time inference.
> It is not true that [1] can be applied to general continual learning such as our experimental settings. It could be possible to train the model of [1] in our continual learning scenario, but it is not possible to apply it at test time since there is no gating mechanism to choose the responsible expert at test time in which no ground truth label is available. In other words, [1] can be task-agnostic only in the settings where class labels are available at test time, such as the continual “few-shot” learning which is the only type of continual learning experiment in [1].
>
> Such limitation of [1] arises from the absence of generative component p(x | φ) when computing responsibility: p(φ | x, y) ∝ p(y | x, φ) p(φ). This formulation of [1] is okay if we have (x, y) pairs at test time such that we can predict the posterior: p(φ | x, y, X, Y) ∝ p(y | x, φ) p(φ | X, Y). However, in the case of discriminative continual learning scenarios (Split-MNIST, Split-CIFAR10/100, …), we need to infer p(φ | x, X, Y) at test time, which is not possible with [1]’s formulation. This is a typical problem of expansion-based continual learning, and we solve it by adding a generative component to the formulation: p(φ | x, y) ∝ p(y | x, φ) p(x | φ) p(φ), p(φ | x, X, Y) ∝ p(x | φ) p(φ | X, Y).
>
> 3. The training procedure is different.
> Although DPM can provide the theoretical basis for task-free continual learning, many additional components are required to stably integrate with neural networks in practice. Therefore, we introduce the wake-sleep cycle with short-term memory (STM), which is crucial to handle a data stream entering one-by-one; otherwise, fitting neural networks to a single data point leads to severe overfitting. On the other hand, [1] does not need a buffer like STM since the experimental setup already assumes that the data comes in as a task (i.e. a set of supports and query examples). We also propose several practical training techniques such as parameter sharing and temperature parameters for the expert classifier.
>
>
> In summary, [1] is a method for continual “few-shot” learning and cannot be applied to general task-free continual learning since it requires task information at test time as many other expansion methods do. Our CN-DPM, on the other hand, is the first expansion method that can be applied to general task-free continual learning.

---

### Author Response · Authors · 2020-01-02
**Camera-ready version is uploaded.**

We thank the anonymous reviewers and the program chair for their constructive and thoughtful comments. We just have uploaded the camera-ready version of our paper and publicly released the code through GitHub (https://github.com/soochan-lee/CN-DPM). We have largely updated section 2 for better comparison with other related papers such as (Nagabandi et al., 2019; Jerfel et al., 2019; Rao et al., 2019).

We also have added appendix M to compare our method with CURL (Rao et al., 2019). To summarize the differences, CURL has some components (the decoder network $p(x|z)$ and the gating network $q(y|x)$) shared across all tasks due to its generative process. As a consequence, expansion alone is not enough to fully prevent catastrophic forgetting, and the generative replay is added to mitigate forgetting in the shared components. Therefore, CURL can be regarded as a hybrid of replay and expansion methods, whereas our CN-DPM is solely expansion-based.

---

### Decision · Program_Chairs · 2019-12-19

**Decision:**

Accept (Poster)

**Comment:**

This paper proposes an expansion-based approach for task-free continual learning, using a Bayesian nonparametric framework (a Dirichlet process mixture model).

It was well-reviewed, with reviewers agreeing that the paper is well-written, the experiments are thorough, and the results are impressive. Another positive is that the code has been released, meaning it’s likely to be reproducible.

The main concern shared among reviewers is the limited novelty of the approach, which I also share. Reviewers all mentioned that the approach itself isn’t novel, but they like the contribution of applying it to task-free continual learning. This wasn’t mentioned, but I’m concerned about the overlap between this approach and CURL (Rao et al 2019) published in NeurIPS 2019, which also deals with task-free continual learning using a generative, nonparametric approach. Could the authors comment on this in their final version?

In sum, it seems that this paper is well-done, with reproducible experiments and impressive results, but limited novelty. Given that reviewers are all satisfied with this, I’m willing to recommend acceptance.